# How Do Gepotidacin and Zoliflodacin Stabilize DNA Cleavage Complexes with Bacterial Type IIA Topoisomerases? 1. Experimental Definition of Metal Binding Sites

**DOI:** 10.3390/ijms252111688

**Published:** 2024-10-30

**Authors:** Harry Morgan, Robert A. Nicholls, Anna J. Warren, Simon E. Ward, Gwyndaf Evans, Fei Long, Garib N. Murshudov, Ramona Duman, Benjamin D. Bax

**Affiliations:** 1Medicines Discovery Institute, Cardiff University, Cardiff CF10 3AT, UK; morganh28@cardiff.ac.uk (H.M.);; 2Diamond Light Source, Harwell Campus, Didcot, Oxfordshire OX11 0DE, UK; 3Scientific Computing Department, UKRI Science and Technology Facilities Council, Harwell Campus, Didcot, Oxfordshire OX11 0DE, UK; robert.nicholls@stfc.ac.uk; 4MRC Laboratory of Molecular Biology, Cambridge CB2 0QH, UK

**Keywords:** novel bacterial topoisomerase inhibitor (NBTI), spirocyclic pyrimidinetrione (SPT), gepotidacim, zoliflodacin, type IIA topoisomerase, DNA cleavage

## Abstract

One of the challenges for experimental structural biology in the 21st century is to see chemical reactions happen. *Staphylococcus aureus* (*S. aureus*) DNA gyrase is a type IIA topoisomerase that can create temporary double-stranded DNA breaks to regulate DNA topology. Drugs, such as gepotidacin, zoliflodacin and the quinolone moxifloxacin, can stabilize these normally transient DNA strand breaks and kill bacteria. Crystal structures of uncleaved DNA with a gepotidacin precursor (2.1 Å GSK2999423) or with doubly cleaved DNA and zoliflodacin (or with its progenitor QPT-1) have been solved in the same P6_1_ space-group (a = b ≈ 93 Å, c ≈ 412 Å). This suggests that it may be possible to observe the two DNA cleavage steps (and two DNA-religation steps) in this P6_1_ space-group. Here, a 2.58 Å anomalous manganese dataset in this crystal form is solved, and four previous crystal structures (1.98 Å, 2.1 Å, 2.5 Å and 2.65 Å) in this crystal form are re-refined to clarify crystal contacts. The structures clearly suggest a single moving metal mechanism—presented in an accompanying (second) paper. A previously published 2.98 Å structure of a yeast topoisomerase II, which has static disorder around a crystallographic twofold axis, was published as containing two metals at one active site. Re-refined coordinates of this 2.98 Å yeast structure are consistent with other type IIA topoisomerase structures in only having one metal ion at each of the two different active sites.

## 1. Introduction

How the two bacterial type IIA topoisomerases, DNA gyrase and topoisomerase IV, cleave DNA is not merely of academic interest. DNA gyrase and its homolog, topoisomerase IV, are targets of well-characterized fluoroquinolone antibiotics (e.g., ciprofloxacin, ofloxacin, levofloxacin and moxifloxacin), which continue to be widely used in clinical practice despite serious safety concerns [1]. Such compounds, known as topoisomerase poisons, stabilize intermediate DNA cleavage complexes formed during catalysis that lead to accumulation of DNA double strand breaks and bacterial cell death [2]. Structural studies explained how common topoisomerase mutations confer clinical resistance to fluoroquinolones [3]. Recently, the first members of two new classes of gyrase-targeting antibiotics, which are not susceptible to these common fluoroquinolone resistance mutations, have successfully completed phase III clinical trials (and could therefore be marketed for clinical use): a spirocyclic pyrimdinetrione (SPT), zoliflodacin [4], and a ‘novel bacterial topoisomerase inhibitor’ (NBTI), gepotidacin [5,6]. Zoliflodacin is derived from its progenitor QPT-1 (for quinoline pyrimidine trione [7]). Zoliflodacin [8] and QPT-1 [9] stabilize double-stranded DNA breaks by sitting in the two, four base-pair separated DNA cleavage sites, in a similar manner to quinolones such as moxifloxacin [3,9,10]. NBTIs such as GSK299423 [11] and gepotidacin [6] have been observed to bind on the twofold axis of complexes, midway between the two four base-pair staggered DNA cleavage sites, and have been observed to stabilize single-stranded DNA cleavage complexes [6,11]. The rise of antibacterial resistance in the 21st century is a major global healthcare threat; the new NBTI and SPT classes of antibacterial drugs should help with this problem.

Antibacterial resistance occurs when bacteria develop mechanisms to survive, grow and multiply in the presence of antibacterial compounds, which are normally capable of killing bacterial cells [12,13]. Studies have shown that infection by antimicrobial-resistant (AMR) pathogens can lead to severe illnesses and in certain cases, death [12,14,15]. In the United States, the Center for Disease Control (CDC) estimated the cost of AMR to be USD 55 billion per year, considering both healthcare costs and productivity loss [15]. Findings from the European Centre for Disease Control (ECDC) showed that across the three main antimicrobial groups presented (carbapenems, fluoroquinolones and aminoglycosides), the increase in the number of antimicrobial-resistant isolates was greater in 2021 when compared to the averages for 2018 and 2019, at an average increase of +121% across the three antimicrobial groups [16]. Although resistance to the bacterial topoisomerase targeting fluoroquinolones (ciprofloxacin/levofloxacin/ofloxacin) was only some 5% to <10% in 2021 in the UK and France, eastern European countries like Poland and Ukraine reported figures of 25% to <50% [16], while in Russia and Turkey, fluoroquinolone-resistant *E. coli* was present in ≥50% of invasive isolates [16]. Such high percentages of *E. coli* resistance to these compounds are concerning, as the Gram-negative *E. coli* is the dominant bacteria responsible for many common human diseases, including, but not limited to, urinary tract infection (UTI), enteritis and septicemia [17]. It is not yet certain if the higher resistance rates seen in some countries are due to the presence of antibiotics in the environment [18]. The NBTI gepotidacin has successfully passed a phase III clinical trial for urinary tract infection (UTI) [19], while zoliflodacin [20] has recently successfully passed a phase III clinical trial for a Gram-negative bacteria-caused infection, gonorrhea [21]. The balanced dual targeting of both DNA gyrase and Topoisomerase IV by gepotidacin decreases the chance of resistance developing [22].

Enzymes which change the topology of DNA, topoisomerases, are divided into two classes, type I and type II, which introduce single-stranded and double-stranded DNA breaks, respectively, to modify the DNA topology [23,24,25]. DNA gyrase is a bacterial type IIA topoisomerase that can regulate DNA topology by making a temporary double-stranded break in one DNA duplex (in the gate or G-segment, forming a so called cleavage complex) and then passing another DNA duplex (the transport or T-segment) through this break; this is accompanied by large movements in the enzyme [23] (see Appendix A). Zoliflodacin [8] and QPT-1 [9] are both capable of trapping the four base-pair staggered break in the DNA, as are quinolones.

An *S. aureus* DNA gyrase P6_1_ crystal system, originally developed to support the development of NBTIs such as gepotidacin [11], has also now produced crystal structures (see Table 1) with both uncleaved DNA [11,26,27,28,29,30,31,32] and with doubly cleaved DNA complexes stabilized by zoliflodacin [8] or its progenitor QPT-1 [9] and an imidazopyrazinone [33], suggesting that, with the proper design of the chemistry [34] of an experiment, it might be possible to observe DNA cleavage taking place in these crystals. Indeed, careful refinement of an imidazopyrazinone complex suggested that one DNA cleavage site contained both cleaved and uncleaved DNA (see supplementary Figure S2 in [33]). Chemistry for pump probe-type reactions is being developed [35], and reactions within crystals normally proceed between 10 and 10,000 times slower than in solution [36]. It may be possible, with careful design of chemistry, to perform such experiments at a synchrotron [37]; however, the possibility of conducting such experiments at an XFEL [37] should not be ruled out.

By the time a paper (submitted in October 2009 with the title ‘Structural basis for DNA cleavage by type IIA topoisomerases revealed by a new class of broad spectrum antibacterial’) was published in August 2010 (entitled ‘Type IIA topoisomerase inhibition by a new class of antibacterial agents’ [11]), the first structure showing the existence of the quinolone water metal ion bridge [3] had been accepted. To indicate the problems encountered in determining an *S. aureus* gyrase 3.35 Å with the quinolone ciprofloxacin structure (pdb code: 2XCT), it was chosen, in 2010, to release coordinates with a clearly incomplete coordination geometry for the quinolone-associated metal ions. Corrected coordinates, consistent with subsequently determined better higher-resolution quinolone structures [3,9,10], are now available from [38] (click on ‘Research’ tab and scroll down, then click on ‘2xct-v2-BA-x.pdb‘ to download coordinates (the Protein Data Bank did not accept deposition of these corrected coordinates)).
ijms-25-11688-t001_Table 1Table 1Twenty published P6_1_ *S. aureus* DNA gyrase GyrB27–A56(GKdel) structures. Four re-refined structures (*-v2-BA-x.pdb*) are highlighted (underlined structures are available with standard BA-x nomenclature from ‘Research’ tab at https://profiles.cardiff.ac.uk/staff/baxb—click to download (accessed on 10 October 2024).No.Ref.PDBCode + Res in ÅInhibitorTypeDNA Sequence Central SixBase-Pairs ^1^Inhibitor PocketsOccupied ^2^Catal. Tyr 123 ^3^



11′2D2A33′
1 [11]**2xcs-v2-BA-x.pdb*** 2.1 ÅNBTI5′ GGGCCC 3′--xx--Phe2 [27]4bul-BA-x.pdb 2.6 ÅNBTI5′ TGTACC 3′3′ ACATGG 5′--xx--Phe3 [29]4plb 2.69 ÅNBTI5′ GGGCCC 3′--xx--Phe4 [28]5bs3 2.65 ÅNBTI5′ GGGCCC 3′--xx--Phe5 [9]**5cdm-v2-BA-x.pdb*** 2.5 ÅSPT5′ C-^Y^GGCCG 3′xx----Tyr^P^6 [9]5cdp-BA-x.pdb 2.45 ÅEtop.5′ G-^P^GTACC 3′x-----Phe7 [9]**5cdr-v2-BA-x.pdb*** 2.65 Å-5′ G-^P^GTACC 3′------Phe8 [26]**5iwi-v2-BA-x.pdb*** 1.98 ÅNBTI5′ GGTAC A 3′3′ CCATG-T 5′--xx--Phe9 [26]5cdp-BA-x.pdb 2.5 ÅNBTI5′ GGTTCA 3′3′ CCAAGT 5′--xx--Phe10 [39]5npp-BA-x.pdb 2.22 ÅNBTI + thiop.5′ G-^P^GTACC 3′--xxxxTyr11 [30]6fm4 2.7 ÅNBTI5′ GGGCCC 3′--xx--Phe12 [33]6fqs-BA-x.pdb 3.11 ÅIPY5′ C-^Y^GGCCG 3′xx----Tyr^P^13 [6]6qtk-BA-x.pdb 2.31 ÅNBTI5′ G-^P^GTACC 3′--xx--Phe14 [40]6qx1-BA-x.pdb 2.65 Åbenzoi’5′ G-^P^GTACC 3′----xxPhe15 [41]6z1a 2.3 ÅNBTI5′ G-^P^GTACC 3′--xx--Phe16 [31]7fvs 2.16 ÅNBTI5′ GGGCCC 3′--xx--Phe17 [31]7fvt 2.08 ÅNBTI5′ GGGCCC 3′--xx--Phe18 [32]7mvs 2.60 ÅNBTI5′ GGGCCC 3′--xx--Phe19 [8]8bp2-BA-x.pdb 2.80 ÅSPT5′ C-^Y^GGCCG 3′xx----Tyr^P^20 t.p.9fz6-BA-x.pdb 2.58 Å-5′ G-^P^GTACC 3′------Phe^1^ Where only a single sequence is given, the two sequences in the duplex are identical. Where the two strands have different DNA sequences, the DNA has static disorder around the twofold axis. ^2^ Pockets are defined in Bax et al. (2019 = [42]). ^3^ In all structures determined to date, the two catalytic residues are the same. If the catalytic Tyr is mutated to a Phe, it cannot cleave the DNA. Residues Tyr^P^ have cleaved the DNA. * = re-refined coordinates in this paper (t.p.)—see Section 4 below.

Also, in 2010, a paper purporting to show ‘*A novel and unified two-metal mechanism for DNA cleavage by type II and IA topoisomerases*’ was published in *Nature* [43]. The mechanism proposed in this paper was based on a 2.98 Å structure in which two metal ions (zinc ions) had been placed in peaks in an electron density map. However, this published structure (pdb code: 3L4K) takes no account of the static disorder around a crystallographic twofold; the two different active sites are modelled with exactly the same metal ion coordination geometry and both active sites ignore normal rules [44,45] of metal ion coordination geometry. Consequently, this structure has incorrect coordination geometry around these two metal ion sites, which does not make chemical sense; it should be corrected [46]. This yeast structure has now been refined consistently with a single moving metal mechanism [42] and with reasonable metal ion coordination geometry at each of the two different active sites (coordinates at [38]; click on research tab and scroll down then click on RR-3l4k.pdb—RR = re-refined). One of the re-refined structures (see Table 1), the 1.98 Å 5iwi-v2-BA-x.pdb, suffers from similar static disorder to the 2.98 Å 3lk4 structure.

Two of the other re-refined [47] P6_1_ crystal structures, the 2.1 Å 2xcs-v2-BA-x.pdb and the 2.5 Å 5cdm-v2-BA-x.pdb, are used in the accompanying paper on a single metal mechanism. Re-refining these two structures did not significantly alter the coordination geometry at the active sites; however, it did clarify an important role played by a partially occupied manganese ion (C 901) and a BisTris buffer molecule (C 902) at a crystal contact (with one end of the DNA and a histidine residue whose protonation state inversely correlates with the occupancy of the manganese ion).

Finally, an *S. aureus* DNA gyrase binary (DNA and protein) crystal structure was also determined, based on data collected at an X-ray wavelength of 1.33 Å (E = 6.6 keV, above the Mn K-edge) (see Section 4 and [48] for details). This structure shows that, in addition to the fully occupied catalytic metals, whose occupancies were defined by ensuring that their temperature factors were comparable to those of the surrounding atoms [49], additional sites are present with lower occupancies. In addition to the expected crystal contact partially occupied manganese site [8], density for additional manganese ions was seen near the quinolone water metal ion bridge in this binary complex with nicked DNA. Consequently, the corresponding 2.65 Å binary structure was also re-refined (5cdr-v2-BA-x.pdb).

In publishing a previous paper on this subject, ‘*DNA Topoisomerase Inhibitors: Trapping a DNA-Cleaving Machine in Motion*’ [42], the ideas now fully developed in the accompanying paper (Nicholls et al., 2024) [50] were placed in the Appendix A. The two biochemists on the paper, Thomas Germe and Tony Maxwell, proposed a scheme, described in the main body of that paper (pp. 3438–3447) [42], which seemed to ignore chemistry ‘rules’ derived from small molecule crystal structures [44,45]. The single moving metal mechanism described in the accompanying paper is more probable (see Section 3). The mechanism described in [50] has sensible chemistry.

## 2. Results

In this paper, our definitions of low- (>3 Å), medium-low- (2.51–3.0 Å), medium-high- (2.01–2.50 Å) and high (2.0 Å or better)-resolution X-ray crystal structures of type IIA topoisomerases are defined by how easy it is to determine the coordination geometry of a metal ion at that resolution. These resolution definitions are similar to those used in structure-guided drug design, where ‘unhappy’ waters (which can be displaced) need to be clearly distinguished from ‘happy’ waters (which have ‘optimal’ contacts) [51]. For parts of structures complicated by static disorder around a twofold axis (often seen with type IIA topoisomerases), the comparatively poor electron density in these regions invokes map interpretability more akin to structures of around 0.5 Å worse resolution. While the initial 2.1 Å crystal structure of GSK299423 had no ambiguity for the catalytic metal ion coordination sphere (see Figure 5 in [11]), the compound, GSK299423, sat on the twofold axis of the complex and was effected by static disorder (see supplementary Figure S10b–d in [11]). This meant that the water structure around GSK299423 was not very clearly defined in the electron density maps, and since individual water molecules can play an important role in ligand recognition [52], we sought to improve this situation. The major aim at that time was to support chemistry aimed at developing new NBTIs, and a better crystal system was sought by trying to exploit the different crystal packing observed on the two ends of the DNA duplex in the P6_1_ crystal system (see Section 2.3). After two years of failure in trying to exploit the slight asymmetry in crystal packing in this crystal form, in 2007, this attempt was abandoned when a structure with GSK945237 and a ‘nicked’ DNA gave a 1.98 Å structure suffering from static disorder around the twofold axis of the complex [26]; the DNA and compound both suffered from static disorder (whereas in the original GSK299423 structure, only the compound suffered from static disorder).

In this paper, the Y and 3′ nomenclature for the two observed metal binding sites (see Figure 1) is used (see Section 4 for details). This was the original nomenclature proposed for the two sites in 2009/2010 [11]. However, the A = 3′ and B = Y site nomenclature is also used in case the reader is more familiar with this naming system for the two metal sites. Having the same names for the single metal ion (5081) and the water molecules involved in catalysis is also helpful (see [50] for details).

### 2.1. High-Resolution Structures Have Clearly Defined Metals at Either the 3′(A)-Site or Y(B)-Site

In vivo type IIA topoisomerases are believed to use Mg^2+^ ions to cleave DNA; however, in vitro, both Mn^2+^ and Ca^2+^ ions can also be used by the enzymes to cleave DNA. The two single-strand DNA cleavage events take place one after the other, resulting in a double-strand break with 4 bp 5′ overhangs [53]. High-resolution structures of type IIA topoisomerases are difficult to obtain, presumably because of the inherent flexibility of the enzyme. However, the *S. aureus* Greek Key deletion construct tends to give better resolution (see Table 1 for twenty P6_1_ structures published with this construct).

Clear structures of the 3′ (or A-site) metal and many structures with Y (or B-site metals) have now been published (see Figure 1; note Molecular figures generated with Pymol [54]). The catalytic tyrosine is believed to be in the tyrosinate form when it cleaves the DNA, becoming covalently attached by a ‘phosphotyrosine’ bond (see Figure 1, and [50] for more details).
Figure 1Comparison of the structure of GSK299423 (2xcs-v2-BA-x.pdb) with that of QPT-1 (5cdm-v2-BA-x.pdb). (**a**) The 2.1 Å crystal structure of GSK299423 includes a single Mn^2+^ ion at each active site at the 3′(A) position (in contact with the 3′ oxygen—red arrow). If the DNA were cleaved, the bond between the 3′ oxygen and the phosphorous atom would be cleaved (see panel (**b**)). For clarity, only one GSK299423 molecule is shown on the twofold axis of the complex. (**b**) In the 2.5 Å QPT-1 structure, a single metal ion is seen at the Y (or B) position. This is called the Y site because it seems likely that the catalytic tyrosine (Y123) comes into this site as a tyrosinate ion (negative charge on terminal oxygen) occupying a similar position to water 5095. The position(s) of the 3′ oxygen (OH) and phosphorous atom are indicated by red and black arrow(s).
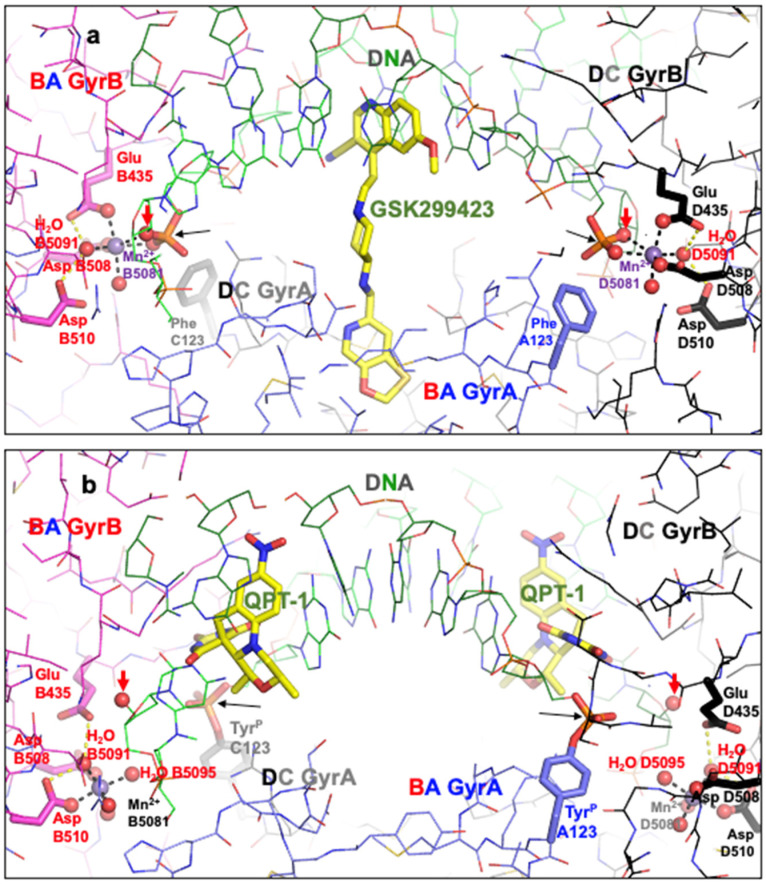


When the catalytic tyrosine has cleaved the DNA (Figure 2), it effectively becomes a phosphotyrosine, and because it was easier to generate crystallographic restraints, it was described as a phosphotyrosine residue (Y^P^123 from A or C chain) in the original 5cdm coordinates. However, the phosphate comes from the DNA, and so in the re-refined coordinates (Figure 1b), tyrosine C123 is covalently linked to E2009, and similarly, tyrosine A123 is covalently bonded to the phosphorous atom of F2009 (see Section 4 for details).

Although both the 2.1 Å GSK299423 structure and the 2.5 Å QPT-1 structure are solved in the same P6_1_ space-group, with the asymmetric unit chosen to be close to the origin, the structures are not in identical positions (see Appendix A). In order to see the movement of the catalytic metal B5081 more clearly, the superposition of structures, as seen in Appendix A, is used in Figure 2. The implied mechanism, with reasonable chemistry, is described in [50] (see Appendix A in this paper for more experimental details).
Figure 2Comparison of metal coordination geometry in the 2.1 Å GSK299423 (2xcs-v2-BA-x.pdb) structure with that in QPT-1 (5cdm-v2-BA-x.pdb). (**a**) The 2.1 Å crystal structure of GSK299423 shows a single Mn^2+^ at the 3′(A) position (in contact with the 3′ oxygen—red arrow). If the DNA were cleaved, the bond between the 3′ oxygen (red arrow) and the phosphorous atom (black arrow) would be cleaved. (**b**) In the 2.5 Å QT-1 structure, a single metal ion is seen at the Y(B) position. This is called the Y site because the catalytic tyrosine (Y123) comes into this site as a tyrosinate ion (-ve charge on terminal oxygen) occupying a similar position to water 5095 for DNA cleavage (see [50]). Oxygens directly coordinating the catalytic metal ion are shown as small spheres, as is the 3′ oxygen (arrowed). (**c**) Panels (**a**,**b**) are shown superposed.
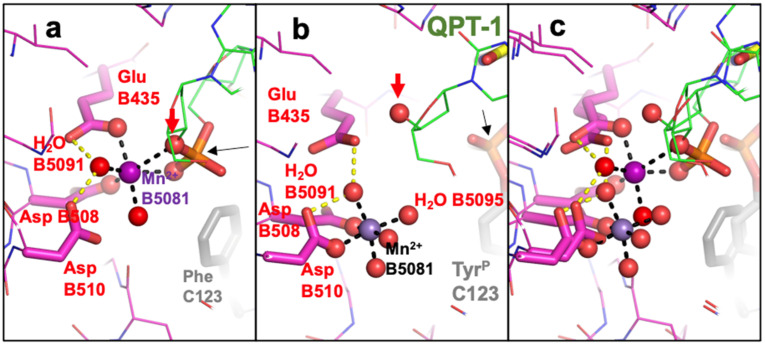


### 2.2. A Comparison of the Original 3l4k Structure Interpreted as Having the Same Metal Ion Coordination Geometry Independent of the Active Sites with Re-Refined Coordinates

In 2010, a structure was published and deposited in the PDB [43] (pdb code: 3L4K). This structure [43] followed some excellent biochemistry from Deweese and Osheroff [55,56,57] which had been interpreted as supporting a two-metal mechanism. However, this interpretation [43] ignored the existence of two active sites in type IIA topoisomerases (see supplementary discussion in [11]). Although the deposited coordinates for 3L4K [43] clearly contain two different active sites, they contain identical metal ions at each active site (see Figure 3a,b). The coordinates clearly imply that, despite having two different active sites in the coordinates, the authors believe that this makes no difference to the metal ion coordination geometry. Because 3L4K sits on a crystallographic twofold axis, the observed 2.98 Å electron density is effectively a convolution of two structures superposed, related by the crystallographic twofold axis. This makes refinement and interpretation of the electron density more challenging, and more ambiguous than would be the case for an otherwise typical 2.98 Å X-ray crystal structure not suffering from such static disorder. Deconvoluted coordinates of the two superposed structures are provided (3lk4-c1a.pdb and 3lk4-c1b.pdb) to clarify the chemistry. In any one yeast dimer, only one of these structures exists, but the crystal is composed of many such structures in random orientations about the twofold axis of the dimer, so the observed electron density is an average. The re-refined coordinates for 3L4K are consistent with a 3′(A)-site (Figure 3c) in the presence of the scissile phosphate and a Y(B)-site metal in the absence of the scissile phosphate (Figure 3d). While the total number of electrons is similar in the two structures (3l4k and re-refined 3l4k—RR-3l4k.pdb, for which we also provide deconvoluted structures: RR-3l4k-c1a.pdb and RR-3l4k-c1b.pdb), the temperature factors for the two fully occupied zincs in the originally deposited coordinates (Figure 3a,b) are much higher (129 and 138) than those of surrounding atoms, while in the re-refined coordinates, the temperature factors (67 and 72) are similar to those of the surrounding atoms [49]. The original models lack waters and have very unusual coordination spheres (Figure 3a,b). The similarity of the re-refined coordinates to those of 5iwi is noted.

### 2.3. Re-Refined P6_1_ S. aureus Structures Suggest a Manganese Ion and BisTris Mediate an Important Crystal Contact

In order to clarify the crystal contacts in the P6_1_ crystal form, we re-refined some crystal structures. This followed on from the determination of 8bp2 [8], a structure with zoliflodacin, which was unusual in being crystallized at a high pH; it was crystallized by microbatch under oil from equal volumes of protein in 342.9 mM HEPES pH 7.2 and precipitate which contained 90 mM Bis-Tris pH 6.3. Usually, the protein comes in 20 mM HEPES. In the 8bp2, a crystal contact manganese ion (named Mn C901) was modelled at full occupancy coordinated by His C390 (see Section 4 for a description of the standard BA-x naming convention used in *S. aureus* DNA gyrase crystal structures). This suggested to us that the variable density seen at this crystal contact might inversely correlate with the protonation state of the histidine C390 side-chain. The re-refined structures seem to confirm this (see Table 2), with a higher occupancy for the pH 6.5 structure (2xcs-v2-BA-x.pdb) than for the pH 6.2 structures. The 5iwi structure suffers from static disorder around the non-crystallographic twofold axis of the complex, in a similar manner to 3l4k. When the 5iwi structure was initially solved in 2007, with one strand with the scissile phosphate in place and the other strand with an artificial nick in the DNA, attempts at getting asymmetric DNA to crystallize in a single orientation in this P6_1_ cell were abandoned [26,27].

### 2.4. An Anomalous Dataset Confirmed Catalytic Metal and Crystal Contact Manganese Ions, but Unexpectedly Showed Additional Manganese Density near Serine 84

The structure (PDB code: 9fz6) was solved from anomalous data collected on beamline I23 [48] at Diamond Light Source (DLS) using 5cdr as the starting model. The structure showed large anomalous peaks (see Table 3) at the two catalytic sites as well as the Mn-coordinated BisTris at a P6_1_ crystal contact (see Table 3 and Figure 4), and unexpectedly, it also showed a new Mn^2+^ binding site close to both the free 5′ phosphates of the 12p DNA on both strands in this binary complex and close to the side-chain hydroxyl of Ser A84 (or Ser C84). In standard BA-x nomenclature, we named this site E or F/1999 (Table 3). An iterative refinement strategy was implemented to estimate Mn^2+^ site occupancies, in which the correlation between the Mn^2+^ B-factors and the B-factors of coordinating residues was optimized [49].
ijms-25-11688-t003_Table 3Table 3Mn^2+^ site occupancies compared to anomalous peak heights.Mn SiteOccupancyPeak Height (rmsd)B/50811.015.5D/50811.015.1E/19990.67.0F/19990.54.3C/9010.24.0Peak height values were calculated with ANODE [58].

Electron density at the crystal contact was modelled with a BisTris molecule, consistent with what has been seen previously in 8BP2 [8] and with other structures (see Table 2). These anomalous data give further evidence that the BisTris is required for the formation of P6_1_ DNA gyrase/DNA crystals (see Appendix A for an initial wrong attempt, in 2xcs, to model this density). In our experience, this P6_1_ crystal form tends to grow as two crystals joined by a merohedral twin plane. The anomalous data were collected from two such crystals; therefore, the crystal structure is modelled as a near perfect twin. Data were collected at two wavelengths (see Section 4), as seen in Figure 5.
Figure 5Experimental anomalous difference Fourier maps. Calculated in *phenix.refine* [59] using data collected at 4.5 keV and 6.6 keV (manganese K-edge), to confirm the presence of manganese and to aid in the refinement of DNA bases. The 4.5 keV map is contoured to 5.5 σ (green) and the 6.6 keV map to 6.5 σ (yellow).
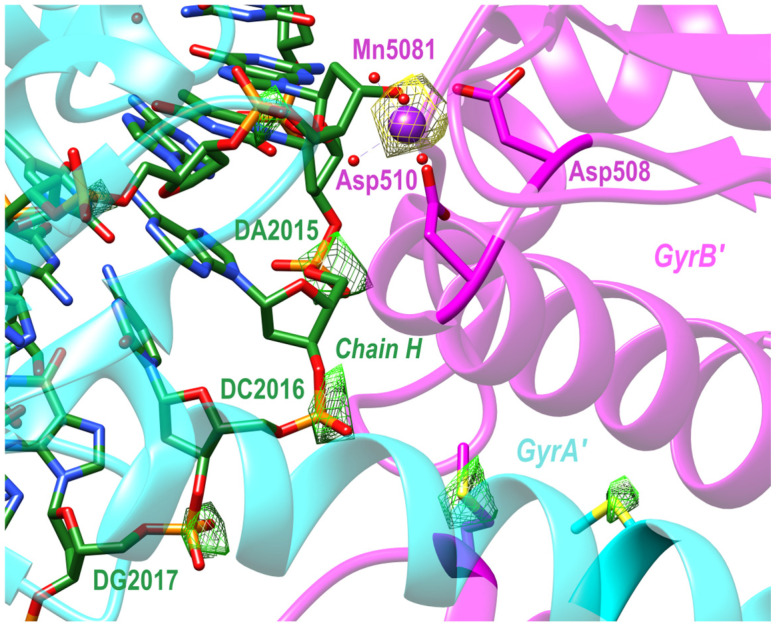


## 3. Discussion

The *S. aureus* DNA gyrase Greek Key deletion construct, which was originally made to support the development of the NBTI gepotidacin [6,11], tends to crystallize in the P6_1_ space-group [42], and crystals in this space-group have also resulted in structural models for QPT-1 [9] and zoliflodacin [8]. Original crystallization screening experiments in 2005 suggested that crystallization only occurs in the presence of BisTris buffer. Because the protein is purified in the presence of 5 mM MnCl_2_, it was proposed in 2023 [8] that observed electron density on one end of the DNA corresponds to a Mn^2+^ ion sitting on His C390 coordinated by BisTris (see Figure 4). With decreasing pH, there is less electron density; this histidine C390 side-chain presumably becomes protonated (see [31] for crystals at pH 5.5, lacking any Mn^2+^ electron density on His C390). However, in our experience this P6_1_ crystal form tends to grow, like back-to-back small pencils, from a merohedral twin plane. It is not clear why this is, but one possibility is that where the two P6_1_ crystals are joined—in opposite orientations—this Bis-Tris manganese ion is required on both His C390 and His A390. As the crystals grow away from the merohedral twin plane, both adopt the same asymmetry but in opposite directions (i.e., His A390 becomes His C390 in one of the two crystals).

This paper has shown how all published type IIA topoisomerase structures are consistent with a single moving metal mechanism. Such a mechanism is presented in an accompanying paper [50], and current estimates of the probability of each of proposed mechanisms being correct are given in Table 4. There are currently three theories (see Table 4) of how the G-segment DNA is cleaved by type IIA topoisomerases such as DNA gyrase: (i) a two-metal mechanism, which was proposed [43] based on what is clearly, to any experienced, technically competent crystallographer (see Figure 3), the misinterpretation of an electron density map [43] (to date, to the best of our knowledge, there have been no structural data published that are consistent with such a model, so it is unlikely to be correct); (ii) a single metal mechanism in which DNA cleavage is proposed to take place when the metal ion ‘appears’ at the 3′(A) site (described in pp. 3438–3447, without a description of how the metal gets to the 3′(A) site) [42]. This model is also unlikely to be correct; it seems to ignore normal rules of coordination geometry [44,45]. The reason this model was published was because it is difficult to prove it is wrong, and the advice in the paper entitled ‘*Detect, correct, retract: How to manage incorrect structural model*’ had not yet been read [46].

Note that, although it is not possible to entirely rule out that the first and second DNA cleavage steps could have different mechanisms, it is difficult to envision how such different mechanisms would have evolved. The first and second DNA cleavage steps seem likely to use similar mechanisms.

The *S. aureus* Greek Key deletion construct has been useful in developing the NBTI class of topoisomerase inhibitors. Indeed, this P6_1_ crystal form (see Table 1 for details) was initially developed to support the development of gepotidacin. In GlaxoSmithKline, the development of the NBTI gepotidacin was problematic because of the general hERG (and other ion channel) inhibition of the class [60]. The hERG liability of the NBTI class can occasionally be overcome by careful screening of compounds [26,27,60]. However, virtually all current NBTIs have a basic nitrogen center flanked by aromatic or hydrophobic groups (Figure 6), which is a generic pharmacophore description of compounds with a hERG liability [61,62]. Figure 6 and the movie (Appendix A) propose a simple model to explain why the basic nitrogen is usually required in NBTIs. This model proposes that the positive charge on the compound is attracted to the phosphate backbone of the DNA, and then the LHS intercalates. Then, when the DNA is bound by *S. aureus* DNA gyrase, the basic nitrogen ‘sees’ and is attracted to the two Asp 83 s (Figure 6 and Appendix A). If the RHS is not appropriately positioned, it will not have time to enter the pocket on the twofold axis between the two GyrA subunits before the enzyme moves to complete the double-stranded DNA cleavage (see also Figure 2 in [11]). One possibility is that for some enzymes, such as *S. aureus* TopoIV, the compound bound to DNA does not see the Asp 83s properly, accounting for weaker activity.

If a robust crystal system could be developed that is suitable for an XChem screen [63], with the pocket between the two GyrA subunits already open, then fragments containing an RHS but lacking the basic nitrogen might be found. One obvious possibility is to replace the basic nitrogen with a ligand-coordinated metal ion; another possibility would be to try and crystallize a complex with the D83N mutant and no RHS on a compound to try to see if this is a realistic proposal.

## 4. Materials and Methods

In the P6_1_ crystal form of the *S. aureus* DNA gyrase fusion truncate (see Table 1), there is one complex in the asymmetric unit.

### 4.1. The Standard BA-x Nomenclature for S. aureus DNA Gyrase Crystal Structures and the Y(B), 3′(A) Nomenclature for Catalytic Metals

In *S. aureus* DNA gyrase^CORE^ structures, the chains are named B (GyrB) and A (GyrA) from the first gyrase^CORE^ fusion truncate subunit, and D (GyrB) and C (GyrA) in the second subunit (we call this nomenclature BA-x for GyrB/GyrA extended numbering; see also supplementary discussion in Bax et al., 2019 [42]). The DNA strands standardly have ChainIDs as E and F and each strand contains a total of twenty nucleotides; however, when the DNA is cleaved, twelve nucleotides from chain E become covalently attached to tyrosine C123. To easily distinguish between amino acid and nucleotide residues, we add 2000 to nucleotide residue numbers, i.e., E2009–E2020 are nucleotides covalently attached to tyrosine C123 after DNA cleavage. Similarly, for DNAs containing artificial nicks at each DNA cleavage site (such as 20-12p-8), the eight nucleotides before the nick are numbered E1–E8 (or F1–F8), while those after the nick are numbered E2009–E2020 (or F2009–F2020). Toprim-bound metals are given the ChainID of the Toprim domain to which they are bound and the number 5081 (i.e., B5081 for catalytic metal coordinated by the B GyrB chain, D5081 for the metal coordinated by the D GyrB chain). This nomenclature is used to indicate that in all *S. aureus* DNA gyrase structures solved to date, Asp 508 (from GyrB) moves to coordinate the metal ion whether it occupies the 3′(A) or Y(B) site. One water molecule, which is clearly conserved between the 3′(A) and Y(B) sites, is numbered 5091, and is called ‘the inside water molecule’. The coordination sphere for the 3′A) site includes only two water molecules, 5091 and 5090 (the second molecule contacts GyrB Asp510 and is displaced by it at the B site). In contrast, the Y(B) site metal coordination sphere has four water molecules, 5091 and 5093, 5094 and 5095. Water 5095 is displaced by the side-chain oxygen of the catalytic tyrosine (A123 or C123 from GyrA) in the model of DNA cleavage presented in the accompanying paper [50].

In this paper, the Y and 3′ nomenclature for the two observed metal binding sites is used because (i) it avoids confusion with B (GyrB) and A (GyrA) ChainIDs, (ii) in structures with partial occupancy, it avoids confusion with the use of alternative position codes A and B, (iii) it seems informative of the mechanism (see [50]), and (iv) it was the original nomenclature proposed in October 2009 for the two sites [11]. However, we also use throughout the A = 3′ and B = Y site nomenclature in case the reader is more familiar with this naming system for the two metal sites. Having the same names for metal ions and waters involved in catalysis is useful in making movies from more than one structure (see [50]).

### 4.2. Re-Refining Fully and Partially Occupied Metal Binding Sites with Chemically Reasonable Geometry

To refine a fully occupied metal, it is desirable to have 2 Å (or better) X-ray data, as this normally allows unambiguous interpretation of ‘heavy atom’ (carbon, nitrogen, oxygen and heavier elements, but not hydrogen atoms) positions. Between 2.01 and 2.50 Å, unambiguous interpretation of the water structure around fully occupied metal ions can become ambiguous (see Appendix A) and restraints on metal ion coordination geometry are usually introduced somewhere in this resolution (between 2.01 and 2.50 Å) range based on an analysis of small molecule crystal structures [44,45]. Lower-resolution structures usually cannot clearly define coordination sphere geometry, and so restraints must be applied for metal ion coordination spheres in refinement (see, for example, Wohlkonig et al., 2010 [3], Pan et al., 2015 [9] and Bax et al., 2019 [42]). Because metal ion site coordination geometry can vary, as seen in small molecule crystal structures [44,45], it is desirable to have clear high-resolution structures that unambiguously define the coordination sphere for use with lower-resolution structures.

With partially occupied metal ions, refinement is more problematic and the coordination sphere usually needs restraints in refinement. The occupancy of the crystal contact Mn^2+^ ions (coordinated by the side-chain of His C390, and mediating a crystal contact via a Bis-Tris buffer molecule—see Figure 4) seems to depend on the protonation state of the histidine 390 side-chain (see Table 2). In the initial deposition of 2xcs (2.1 Å GSK299423 structure), there was an error in interpreting the electron density at this important crystal contact (see Appendix A for initially deposited 2xcs model). Periodic Bond Chain theory [64] is often applied to crystal growth [65]. However, most crystals do not grow as a pair of merohedral twins from a common plane perpendicular to the 6_1_ axis; the initiation step from this common plane may require both BisTris and Mn^2+^ ions for the initiation of the growth of this P6_1_ crystal form (Table 1) of the *S. aureus* fusion truncate in complexes with DNA even in the presence of crystal seeds.

The structure factors for four structures we re-refined (PDB codes: 2xcs, 5cdm, 5iwi, 5cdr) were retrieved from the protein data bank (PDB) and converted to .mtz files with the CCP4 (version CCP4 8.0.016) program cif2mtz [66]. These four structures were re-refined (see Appendix A) to correct a crystal contact error; this confirmed the accuracy of the catalytic metal ion coordination geometry in these crystal structures. To confirm the presence of manganese ions at the crystal contact site, an anomalous dataset was collected, which showed the expected anomalous peaks for the two catalytic metals and a weaker peak for the partially occupied crystal contact metal ion (see Figure 4). However, it unexpectedly also showed anomalous peaks close to the position seen in quinolone complexes (see Table 3). The anomalous dataset was collected from a binary complex with a doubly nicked twenty base-pair DNA duplex (20-12p-8); the 5′-phosphate on the 12mer [67] was thought to be responsible for the observation of partially occupied manganese sites close to those observed in the water metal ion bridge in the moxifloxacin structure [9]. The 2.65 Å 5cdr model with the same doubly nicked 20-12p-8 was re-refined to include partially occupied Mn^2+^ ions E1999 and F1999 (5cdr-v2-BA-x.pdb).

The P6_1_ structures re-refined (see Appendix A) include the 2.1 Å structure with GSK299423 (pdb code: 2xcs), the highest-resolution unambiguous 3′(A) site structure (Figure 1). This was re-refined without restraints on Mn^2+^ geometry. Although the 2.5 Å QPT-1 structure (pdb code: 5cdm) clearly has ‘the same’ Y(B) coordination geometry as in the 2.16 Å human top IIb structure with etoposide and magnesium ions [68] (see Appendix A), restraints were needed on the metal ion geometry in re-refining this 2.5 Å QPT-1 structure. Restraints were also used in re-refining the catalytic metals in 5iwi. a 1.98 Å structure in which the catalytic metal ions (Mn^2+^) are complicated by static disorder around the twofold axis, analogous to that seen in 3l4k.

The 5cdr structure was re-refined because the 2.58 Å anomalous dataset, collected at the manganese K edge (6.6 keV = 1.8785 Å), was collected on a crystal grown in similar conditions to 5cdr. Electron density, previously interpreted (in 5cdr) as a glycerol molecule, is now interpreted as a partially occupied Mn^2+^ ion and coordinating waters (5cdr-v2-BA-x.pdb). The lack of high-resolution fully occupied Mn^2+^ ions for this site (near Ser 84) means that this interpretation is not definitive.

Refinement of a protein structure is never complete; it is only stopped when time and publication dictate. The details of re-refined coordinates, re-refined with Refmac5/Servalcat [47,69], are described in Appendix A. In normal protein crystal refinement, a solvent correction is applied which assumes that the non-defined parts of the structure have the same solvent electron density value. It is noted that for the *S. aureus* DNA gyrase crystals used in this paper, most crystals are grown in an excess of DNA; although there is no clear electron density for this DNA, it may be present as the T-segment in disordered solvent regions. This could account for higher R-factors seen with low-resolution (~40–20 Å) data. In our experience, the low-resolution shells of data usually give R-factors with normal protein crystals, after solvent correction, of less than 20%; with DNA gyrase, the R-factors in these low-resolution shells (between 40 Å and 20 Å) tend to be between 20 and 30%. For this reason, a low-resolution cutoff of 18 or 20 Å is usually employed in refinement (see Appendix A). The solvent region, in DNA gyrase crystals grown in an excess of 20mer duplex DNA, is not necessarily of uniform electron density. 

### 4.3. Protein Purification and Crystallization of a DNA Gyrase—DNA Complex

The *S. aureus* DNA gyrase fusion truncate B27:A56 Y123F (GKdel) (M_W_ 78,020) was purchased from Inspiralis. It was expressed in *E. coli* and purified according to the protocol described in [11] with modifications as described [8]. The purified protein was concentrated to 12 mg/mL and stored in a storage buffer (20 mM HEPES pH 7.0, 5 mM MnCl_2_ and 100 mM Na_2_SO_4_). The DNA oligonucleotide used in crystallizations for anomalous data collection, 20-12p-8, was from Eurogentec (Seraing, Belgium). The lyophilized DNA was resuspended in nuclease-free dH_2_O and annealed from 86 °C to 24 °C over 45 min to give an artificially nicked DNA duplex at a concentration of 2 mM.

Crystallization complexes were formed by mixing protein, HEPES buffer and DNA, then incubating on ice for 1 h. Crystals of *S. aureus* DNA GyrB27:A56 Y123F with 20-12p-8 DNA (binary complexes) were grown by the microbatch under oil method, with streak seeding from a seed stock. Seed stock was produced by crushing 3 hexagonal rod-shaped *S. aureus* GyrB27:A56 Y123F (GKdel)/20-12p-8 crystals in 50 μL of crystallization buffer. A crystallization screen consisting of Bis-Tris buffer pH 5.9–6.4 (90, 150 mM) and PEG 5kMME (7–14%) was made. For a single drop, 1 μL of the complex mixture was mixed with 1 μL of crystallization buffer in a 72-well Terasaki microbatch plate, streak-seeded with a whisker dipped in the seed stock, then covered in 2–3 mL of paraffin oil. The plates were incubated at 20 °C and crystal growth was observed between 1 and 14 days. The crystal which gave the 2.58 Å dataset was grown from a crystallization drop consisting of 1 μL protein mixture (0.0424 mM *S. aureus* DNA GyrB27:A56 Y123F, 0.122 mM 20-12p-8 DNA duplex, 2.755 mM MnCl_2_ and 20 mM HEPES) with 1 μL crystallization buffer (150 mM BisTris pH 6.1 and 12% PEG 5kMME). A single hexagonal rod-shaped crystal was transferred to a cryobuffer (15% glycerol, 19% PEG 5kMME and 150 mM Bis-Tris) before flash-cooling and storage in liquid nitrogen for data collection.

### 4.4. Data Collection, Structure Determination and Refinement

The presence of manganese (Mn) was confirmed by taking a fluorescence spectrum. Data were collected on the long-wavelength beamline I23 [70] at Diamond Light Source (Oxfordshire, UK) at two wavelengths: 1.33 Å (E = 6.6 keV, above the Mn K-edge) and 2.75 Å (E = 4.5 keV). The shorter wavelength (λ = 1.33 Å) was chosen to maximize the anomalous signal from Mn, while the aim for the longer wavelength (λ = 2.75 Å) was to provide increased anomalous scattering from the phosphorus atoms on the DNA backbone [48], to aid model building in sections of poor-quality electron density.

Three 360° datasets were measured alternatively, at each energy, using the kappa goniometry to change the orientation of the crystal and increase the multiplicity of the data. The l = 1.33 Å and the l = 2.75 Å data were collected using 100% and 55% beam transmission, respectively, with an exposure of 0.1 s per 0.1° rotation. Data integration and scaling was performed with autoPROC [71], using a I/σI cut-off > 1.0 and CC_1/2_ > 0.3 (Appendix A). The structure was refined with refmac [47,69] starting from a structurally similar model (Appendix A—for refinement statistics. Note the Cruickshank DPI (Å) was calculated by the Online_DPI server [72]) containing the same *S. aureus* DNA GyrB27:A56 Y123F protein and 20-12p-8 artificially nicked DNA (PDB code: 5cdr). As the data were severely twinned (twin fraction = 0.51, twin law = -K,-H,-L), twin refinement was necessary and restraints for bond length (2.17 Å) and bond angles (90, 180°) were used to accommodate an octahedral Mn^2+^ coordination geometry for the five Mn sites. Model building was performed with Coot [73]. As with most previous *S. aureus* DNA gyrase structures without compound (binary complexes with doubly nicked DNAs), there was considerable difficulty in building the four central DNA base pairs bridging the DNA cleavage sites. The final refined model was used for f″ refinement in Phenix [59], with only ‘anomalous groups’ refinement switched on, starting from an f″ value of 0. Anomalous difference Fourier maps were generated in Phenix.

### 4.5. The Generation of a Movie of Geoptidacin Binding

The movie used two published crystal structures of the *S. aureus* fusion truncate, a 2.37 Å gepotidacin crystal structure with uncleaved DNA (6qtp-BA-x.pdb in P2_1_, a = 86, b = 124, c = 94 Å; a = 90, b = 117, g = 90°) and a 2.6 Å binary complex with uncleaved DNA (6fqv-BA-x.pdb in P2_1_, a = 93, b = 125, c = 155 Å; a = 90, b = 96, g = 90°). Briefly, the movie was made in a manner similar to those described [50,74], by taking the DNA from the crystal structure with geoptidacin, leaving the LHS in place and ‘torring’ around freely rotatable bonds to give a structure with DNA similar to that in Figure 6b. The protein from the binary complex was then introduced, the basic nitrogen on the compound (Figure 6a) was attracted to interact with both Asp 83 s, before the pocket between the two GyrA subunits opened up and the compound moved into its ‘final’ position (as observed in the crystal structure of gepotidacin with uncleaved DNA—Figure 6c).

## Figures and Tables

**Figure 3 ijms-25-11688-f003:**
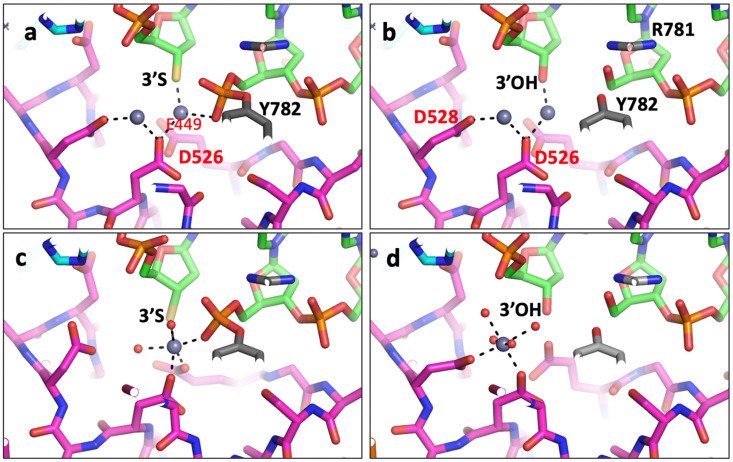
Comparison of metal coordination geometry in the 2.98 Å deposited 3l4k with that in re-refined 3l4k. (**a**,**b**) The two active sites in the originally deposited coordinates are different (occupancies 0.5) but the deposited metals have occupancy 1.0, implying that the authors believe that the metal geometry is unaffected by the presence or absence of the scissile phosphate at the active site. (**c**,**d**) In the re-refined coordinates, the presence of the scissile phosphate at the active site (in (**c**)) attracts a metal ion (zinc) which is octahedrally coordinated by an oxygen from the scissile phosphate, the 3′ sulfur (3′S), oxygens from E449 and D526 and two waters. This metal is at the 3′ site. In (**d**), the absence of the scissile phosphate means that the metal ion is seen at the Y(B) position.

**Figure 4 ijms-25-11688-f004:**
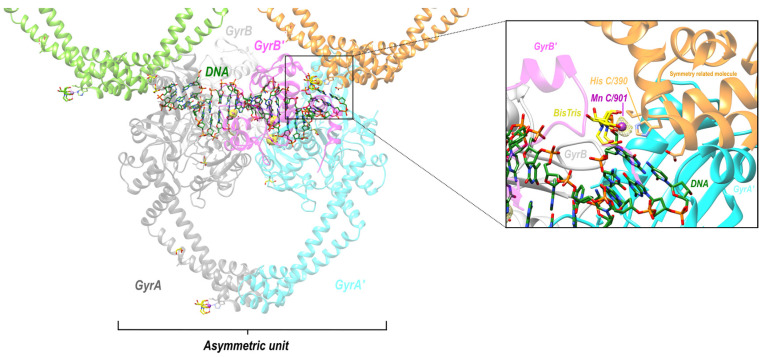
Structure and map at 2.58 Å. In one asymmetric unit, 5 × Mn^2+^ were seen in the f″ refined anomalous difference Fourier map (contoured at 4 σ, shown in yellow; note that the density for the second catalytic metal is obscured by protein structure in this view). Inset anomalous density for BisTris coordinated Mn^2+^ bridging between two asymmetric units at a P6_1_ crystal contact.

**Figure 6 ijms-25-11688-f006:**
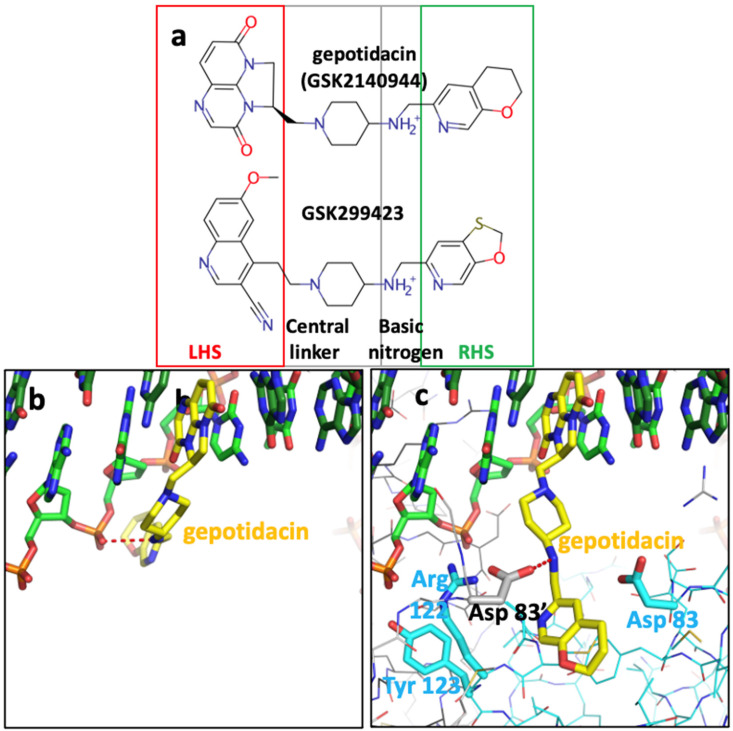
A model of gepotidacin binding to DNA before forming the complex with *S. aureus* DNA-gyrase. (**a**) Gepotidacin and GSK299423 structures compared. LHS—left-hand side; RHS—right-hand side. (**b**) A model of gepotidacin (yellow carbons) bound to DNA. The basic nitrogen has been modelled interacting (dotted red line) with a phosphate of the DNA backbone. (**c**) Gepotidacin bound to *S. aureus* DNA gyrase and DNA; the basic nitrogen interacts with one of the Asp 83 s (dotted red line).

**Table 2 ijms-25-11688-t002:** pH versus Xtal contact Mn occupancy in four P6_1_ *S. aureus* DNA gyrase structures.

PDB	Compound	Resol. (Å)	Crystallization Buffer pH	Mn C901 Occupancy (Bfactor)	His C390 Side-Chain Bfactor
2xcs-v2-BA-x.pdb	GSK299423	2.10	6.5	0.65 (23.3)	23.7
5iwi-v2-BA-x.pdb *	GSK945237	1.98	6.2	0.30 (35.8)	35.5
5cdm-v2-BA-x.pdb	QPT-1	2.50	6.2	0.40 (55.2)	52.1 **
5cdr-v2-BA-x.pdb	-	2.65	6.2	0.30 (37.5)	45.8

* 5iwi-v2-BA-x.pdb contains two DNA strands with different sequences. The relatively high B-factor for the side-chain of His 390 in 5iwi-v2-BA-x.pdb is thought to be due to the two different strands of DNA in 5iwi. ** In 5cdm-v2-BA-x.pdb, the side-chain of histidine 390 was modelled in two conformations, and the B-factors given correspond to that with occupancy 0.4.

**Table 4 ijms-25-11688-t004:** A comparison of models proposed in literature for the first DNA cleavage step of a type IIA topoisomerase.

Mechanism Number and Name	Problems with Mechanism (Current Estimate of Mechanism Being Broadly Correct). Reference.
(i) a two-metal mechanism from 2010 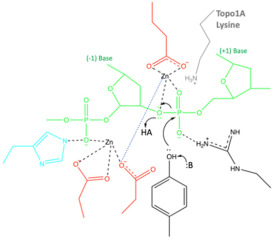	No chemically sensible structure has yet been deposited in support of this mechanism. The X-ray data are consistent with a single moving metal mechanism (see Figure 3). The topo1A lysine is also consistent with the single moving metal mechanism in [50]. Some variation on this two-metal mechanism cannot be entirely excluded: (Low to very low) Schmidt et al. (2010) [43].
(ii) Single-metal mechanism from 2019 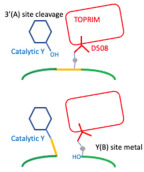	The mechanism proposed does not detail any sensible chemistry. The mechanism seems to propose that the metal starts at the A-site and cleaves the DNA before moving to the B-site. Some variation on this mechanism cannot be excluded: (Low to very low) Bax et al. (2019) [42].
(iii) a single moving metal mechanism described in [50] 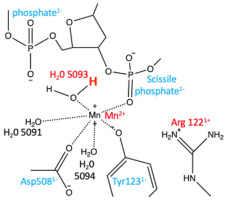	Only heavy atom (non-hydrogen atom) positions defined by experimental data. Mechanism suggests the red hydrogen atom from water 5093 will protonate the 3′ oxygen on the scissile phosphate to cleave the DNA, after which the metaphosphate-like intermediate will be accepted by the catalytic tyrosinate. (High to very high) (Nicholls et al., 2024) [50].
(iv) Some unthought-of mechanism	(Low?). No reference.

## Data Availability

Structure factors and ‘original’ coordinates with standard PDB nomenclature are available from the protein databank (PDB), including the those for 9fz6. Coordinates with standard BA-x nomenclature are available at https://profiles.cardiff.ac.uk/staff/baxb—click on ‘research’ tab and scroll down to download ‘required’ coordinates (accessed on 10 October 2024).

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
