# Peer review of "How Do Gepotidacin and Zoliflodacin Stabilize DNA Cleavage Complexes with Bacterial Type IIA Topoisomerases? 1. Experimental Definition of Metal Binding Sites"

_ijms, 2024, doi:10.3390/ijms252111688_

Round 1

Reviewer 1 Report (Previous Reviewer 1)

Comments and Suggestions for Authors

In this resubmitted manuscript, the authors have re-refined four crystal structures with incorrect metal ion coordination geometry to clarify the crystal contacts and support the proposed mechanism of topoisomerase-DNA cleavage by a single moving metal. In the newly submitted version little more is explained about what was wrong with the four original crystal structures and how the newly refined structures point to the single metal model. While the manuscript is backed up with a large number of references supporting the crystal structure refinement results, the problem I have is that the description of the mechanism of a single moving metal is explained in detail in accompanying paper by Nicholls et al. (2024), which is referenced a lot in this manuscript, but I could not find it. Therefore, I could not verify the parts that refer to this accompanying article.

Author Response

Reviewer 2 Report (Previous Reviewer 2)

Comments and Suggestions for Authors

The manuscript entitled “How do gepotidacin and zolifodacin stabilize DNA-cleavage complexes with bacterial type IIA topoisomerases? 1. Experimental definition of metal binding sites” offers intriguing insights into the crystal structure interpretation of numerous published DNA gyrase-inhibitor complexes.

The search for new antibacterials and the elucidation of their novel mechanisms of action are crucial for innovative antibacterial drug design; thus, the scope of this paper is very interesting.

However, the authors frequently cite Nicholls et al., 2024, and many critical research results are presented in that paper. However, since this paper has not yet been published, reviewers do not have access to it. Consequently, it is challenging to review a manuscript lacking important data that has not been made available. Please, cite the published paper. 

Round 2

Reviewer 1 Report (Previous Reviewer 1)

Comments and Suggestions for Authors

I have no further comments.

This manuscript is a resubmission of an earlier submission. The following is a list of the peer review reports and author responses from that submission.

Round 1

Reviewer 1 Report

Comments and Suggestions for Authors

In the present manuscript, the authors have re-refined four previous crystal structures to clarify the crystal contacts to support the proposed mechanism of topoisomerase DNA cleavage by a single moving metal, which is explained in their second accompanying paper (Nicholls et al., 2024). The manuscript is well written, the study was well conducted and backed up with experiments that suggest valid results. I don't really have any comments, apart from wondering why these particular four (2xcs, 5cdm, 5cdr and 5iwi) crystal structures were chosen for the re-refinement?

Reviewer 2 Report

Comments and Suggestions for Authors

The manuscript entitled “How do gepotidacin and zolifodacin stabilize DNA-cleavage complexes with bacterial type IIA topoisomerases? 1. Experimental definition of metal binding sites” offers intriguing insights into the crystal structure interpretation of numerous published DNA gyrase-inhibitor complexes.

The search for new antibacterials and the elucidation of their novel mechanisms of action are crucial for innovative antibacterial drug design; thus, the scope of this paper is compelling. However, several significant issues must be addressed before its publication in the International Journal of Molecular Sciences.

The authors frequently cite Nicholls et al., 2024, and many critical research results are presented in that paper. However, since this paper has not yet been published, reviewers do not have access to it. Consequently, it is challenging to review a manuscript lacking important data that has not been made available. Therefore, I strongly encourage the authors to first publish Nicholls et al., 2024, and then resubmit their manuscript.